Morphological, histological and transcriptomic mechanisms underlying different fruit shapes in Capsicum spp.

Wang Yixin 1 2
Ma Shijie 1 2
Cao Xiaomeng 1 2
Li Zixiong 1 2
Pan Bingqing 1 2
Song Yingying 1 2
http://orcid.org/0000-0002-0587-2399 Wang Qian 1 2
Shen Huolin 1 2 shl1606@cau.edu.cn
Sun Liang 1 2 liang_sun@cau.edu.cn
1 College of Horticulture, China Agricultural University , Beijing , China
2 Beijing Key Laboratory of Growth and Developmental Regulation for Protected Vegetable Crops, Department of Vegetable Science, College of Horticulture, China Agricultural University , Beijing , China
Nogueira Fabio
Electronic publication date: 2024 Sep 30
Publication date: 2024
Volume: 12
Electronic Location ID: e17909
Received 2023 May 17; Accepted 2024 Jul 22
Copyright: © 2024 Wang et al.
Copyright year: 2024
Copyright holder: Wang et al.
License: This is an open access article distributed under the terms of the Creative Commons Attribution License, which permits unrestricted use, distribution, reproduction and adaptation in any medium and for any purpose provided that it is properly attributed. For attribution, the original author(s), title, publication source (PeerJ) and either DOI or URL of the article must be cited.
License URL: https://creativecommons.org/licenses/by/4.0/

Keywords: Pepper (Capsicum spp.), Fruit shape, Helical growth, RNA-seq, IQD, Calmodulin

Funding: Major Science and Technology plan of Hainan Province ZDKJ2021010 National Natural Science Foundation of China 32372726 Sanya Yazhou Bay Science and Technology City SYND-2021-19, SYND-2022-25 Construction of Beijing Science and Technology Innovation and Service Capacity in Top Subjects CEFFPXM2019_014207_000032 This research was supported by the Major Science and Technology plan of Hainan Province (ZDKJ2021010), the National Natural Science Foundation of China (32372726), the Sanya Yazhou Bay Science and Technology City (SYND-2021-19, SYND-2022-25), and the Construction of Beijing Science and Technology Innovation and Service Capacity in Top Subjects (CEFFPXM2019_014207_000032). The funders had no role in study design, data collection and analysis, decision to publish, or preparation of the manuscript.

==============================
Pepper (Capsicum spp.) has a long domestication history and has accumulated diverse fruit shape variations. The illustration of the mechanisms underlying different fruit shape is not only important for clarifying the regulation of pepper fruit development but also critical for fully understanding the plant organ morphogenesis. Thus, in this study, morphological, histological and transcriptional investigations have been performed on pepper accessions bearing fruits with five types of shapes. From the results it can be presumed that pepper fruit shape was determined during the developmental processes before and after anthesis, and the anthesis was a critical developmental stage for fruit shape determination. Ovary shape index variations of the studied accessions were mainly due to cell number alterations, while, fruit shape index variations were mainly attributed to the cell division and cell expansion variations. As to the ovary wall thickness and pericarp thickness, they were regulated by both cell division in the abaxial-adaxial direction and cell expansion in the proximal-distal and medio-lateral directions. Transcriptional analysis discovered that the OFP-TRM and IQD-CaM pathways may be involved in the regulation of the slender fruit shape and the largest ovary wall cell number in the blocky-shaped accession can be attributed to the higher expression of CYP735A1, which may lead to an increased cytokinin level. Genes related to development, cell proliferation/division, cytoskeleton, and cell wall may also contribute to the regulation of helical growth in pepper. The insights gained from this study are valuable for further investigations into pepper fruit shape development.

Introduction

Pepper (Capsicum spp.) is a popular vegetable that is grown worldwide, and during its long period of domestication and breeding processes, diverse fruit shape variations have been accumulated. Fruit shape is an important agronomic trait for pepper, which not only impacts consumers’ decision, but also determines the use of a certain variety (Sun et al., 2015). For example, small bullet-shaped chili pepper is often used for making pickle; meanwhile, big bell-shaped sweet pepper is popular for making salad; as to the helical-shaped pepper, it is widely grown in Northwest China for stir-frying. In another aspect, fruit shape variation is also a valuable source for illustrating the domestication history of pepper. Thus, clarifying the regulatory mechanism of pepper fruit shape is not only important for deepening our understand of plant organ morphogenesis, but also very useful for pepper breeding.

Fruit shape is considered as a quantitative trait which is regulated by multiple QTLs (quantitative trait loci) (Paran & van der Knaap, 2007). So far, great progresses have been made in another member of the Solanaceae family, tomato, in which six loci, including sun, ovate, sov1, fs8.1, fas and lc, have been fine-mapped (van der Knaap & Østergaard, 2018; Snouffer, Kraus & van der Knaap, 2020). sun locates on the short arm of chromosome 7 and is caused by retrotransposon-mediated insertion of a large segment, which enhanced the expression of IQD12 (SUN) and finally resulted in the elongated fruit through simultaneously up- and down-regulating cell number in proximal-distal and medio-lateral directions, respectively (Xiao et al., 2008; Wu et al., 2011). At molecular level, IQDs are considered as Ca2+-regulated scaffolds which rearrange the structure of cytoskeleton through recruiting CaM, KLCR1 (Kinesin light chain-related protein-1) and SPR2 (SPIRAL2) to the microtubules (Abel, Bürstenbinder & Müller, 2013; Bürstenbinder et al., 2013). ovate is a null mutation on chromosome 2 and is caused by a premature stop codon, which can be attributed to a SNP (Liu et al., 2002). OVATE encodes a member of OFP family and loss-of-function mutation of this gene leads to elongated pear-shaped fruit in the absence of sov1 through promoting cell proliferation in proximo-distal direction and inhibiting cell division in medio-lateral direction, especially at the proximal end of fruit (Wu et al., 2018). sov1 (suppressor of ovate) is resulted from a ~31-Kb deletion in the upstream of SlOFP20, which enhances the expression of SlOFP20 and eventually leads to ovary cell number variation by influencing the cell division pattern (Wu et al., 2018). The functions of the OFP family proteins depend on the interactions between OFPs and TRMs (tonneau 1 recruiting motif protein), which influence the dynamic balance between cytoplasmic- and microtubular-localized OFP-TRM protein complexes and later affect the assembling of the TTP (TON1-TRM-PP2A) complex. The changes in the assembling of the TTP complex impact the organization of microtubule arrays as well as PPB formation, and finally alter cell division patterns and cell growth (Wu et al., 2018). Besides tomato, OFPs have also been proved to be the genes or candidates to control fruit or tuber shapes of other horticultural crops (Ai et al., 2022; Martínez-Martínez et al., 2022; Wang et al., 2022; Wu et al., 2022; Feng et al., 2023), suggesting this mechanism may be conserved and widespread. fs8.1 is fine-mapped to a ~3.03-Mb region on chromosome 8, which regulates fruit shape by mainly increasing cell number in proximal-distal direction (Sun et al., 2015). fas and lc increase locule number and lead to flat fruits. fas has been mapped to chromosome 11 and is caused by a 294-kb inversion with breakpoints in the first intron of YABBY and 1 kb upstream of SlCLV3 start codon (Xu et al., 2015). lc locates on chromosome 2 and is composed of two SNP mutations downstream of the 3′UTR of SlWUS (WUSCHEL) (Muños et al., 2011; Chu et al., 2019). fas and lc play roles in the regulation of tomato fruit shape through impacting the WUS-CLAVATA1 (CLV1) -CLAVATA3 (CLV3) pathway, which is involved in the control of meristem size in Arabidopsis and rice (Schoof et al., 2000; Yadav et al., 2011; Bommert et al., 2013; Perales et al., 2016). Recently, a new gene ENO encoding an AP2/ERF transcription factor has been reported to participate in the CLV-WUS pathway by interacting with the promoter region of SlWUS and finally control tomato fruit locule number and size (Yuste-Lisbona et al., 2020). GLOBE is the gene controlling flat and globe fruit shapes, located on the upper arm of chromosome 12. Knocking out of this gene turns the fruit shape from flat into globe (Sierra-Orozco et al., 2021).

Compared with tomato, pepper has a greater variability in fruit shape of which the classification criteria have not been uniformed. According to different descriptors for Capsicum, pepper fruit shape can be divided into as many as 13 categories, such as flat lantern-shaped (flat-shaped), square lantern-shaped (blocky-shaped), short cattle horn-shaped, string-shaped, etc. Based on the genetic distance which was calculated with SNPs obtained from the re-sequencing of 35 different C. annuum lines, the transitions in fruit shapes were high-likely from round or oval shaped followed by the linear-shaped, short horn-shaped, long horn-shaped, finally to blocky-shaped (Du et al., 2019). In another aspect, many efforts have been made on the clarification of the genetic mechanism underlying pepper fruit shape, and QTLs have been identified on chromosomes (hereafter chr) 1, 2, 3, 4, 6, 8, 10, 11 and 12 (Chaim et al., 2001, 2003; Rao et al., 2003; Zygier et al., 2005; Barchi et al., 2009; Yarnes et al., 2013; Chunthawodtiporn, 2016; Han et al., 2016; Du et al., 2019). Among those QTLs, loci on chr 3 had the largest effects on fruit shape and could explain 14–67% of the total fruit shape variation in the F2, RILs or BC populations derived from intra- or inter-specific crosses between elongated- or oval-shaped accessions and blocky-shaped accessions (Chaim et al., 2001; Rao et al., 2003; Barchi et al., 2009; Han et al., 2016). In addition, a fruit shape locus, fs10.1, explaining 30% of the variation, was mapped on chr 10 in a C. annuum ‘5226’ (spherical-shaped, FSI = 0.7) × C. chinense ‘PI159234’ (short figure-shaped, FSI = 4.9) F2 population, and TG63 was the most significant associated marker (Chaim et al., 2003). Morphological and histological analyses revealed that both fs10.1 and fs3.1 control fruit elongation after anthesis, and fs10.1 mainly regulated fruit shape through changing cell shape in the first 2 weeks of fruit development (Borovsky & Paran, 2011). More recently, an OFP20 gene which locates within the fs10.1 region was confirmed to control pepper fruit elongation (Borovsky et al., 2022). In another study, fruit shape QTLs were identified on chr 2 and 4 in the populations derived from inter-specific crosses. Among those loci, fs2.1 was mapped at the marker ovate in a BC2S1 population, which explained 18.5% of the variation; meanwhile, fs4.1 and fs4.2 were located on chr 4 in the F2 population of IL-315, explaining 16.5% and 26.1% of the fruit shape variation. respectively (Zygier et al., 2005). By using the target SNP-seq genotyping method, nine fruit shape QTLs were mapped on chr 1 (CaSSR013), 2 (CaSSR090), 3 (CaSSR105), 4 (CaSSR091 and CaSSR039), 6 (CaSSR044 and CaSSR107) and 12 (CaSSR077 and CaSNP112), among which the association regions of CaSSR105, CaSSR090, CaSSR044 and CaSNP112 harbored the genes LONGIFOLIA 1-LIKE, OVATE, OFP5 and OFP13, respectively (Du et al., 2019), and member of OFP has already been confirmed to involve in the determination of pepper fruit shape by VIGS (Tsaballa et al., 2011). Recently, transcriptome- and proteome-wide association analysis of 148 RILs, which were derived from a cross between a solitary pod pepper and a sweet pepper, identified two significant QTLs for fruit length (qFL3.1 and qFL7.1) and four QTLs for fruit width (qFWD2.1, qFWD3.1, qFWD6.1 and qFWD11.1) (Liu et al., 2022). Since the region of qRL3.1 almost overlapped with that of qFWD3.1, it can be presumed that a fruit shape QTL is high-likely in the region (Liu et al., 2022). Through resequencing of 311 C. annuum and GWAS, a more recently study clarified the domestication history of the narrow and blocky peppers, and fs-8, fs10.1B and fs11.4 were considered to be the major fruit shape QTLs involving in those processes (Cao et al., 2022b). Additionally, TRM25 and GRP (glycine rich cell wall structure protein) were considered to be the candidates controlling elongated fruit shape and blocky fruit traits, respectively (Cao et al., 2022b). Besides the above-mentioned common fruit shapes, special fruit shapes are also discovered in pepper, among which helical shape is very popular in some parts of Asia. Helical growth is a widespread developmental pattern in plant, which is thought to be directly related to the abnormal arrangement of cytoskeleton and cellulose microfibrils (Buschmann & Borchers, 2020). Mutations of genes encoding α-tubulin protein and microtubule binding/associated proteins have been confirmed to be the reasons underlying helical growth (Buschmann & Borchers, 2020). However, histological and transcriptional mechanisms of the helical as well as other fruit shapes have not been fully illustrated in pepper yet.

Therefore, in this study, morphological, histological and transcriptomic analyses were performed on pepper accessions bearing five types of fruit shape. Results obtained here are not only important for clarifying the regulation of pepper fruit development but also critical for fully understanding the plant organ morphogenesis.

Materials and Methods

Plant materials

Ten pepper (Capsicum spp.) accessions bearing flat-shaped (19C302 and 19C304), blocky-shaped (19C335 and 19C355), horn-shaped (19C961 and 19C1163), helical-shaped (19C616 and 19C705) or slender-shaped (19C1477 and 19C1511) fruits (Fig. 1A) were grown in a standard greenhouse at the Experimental Station of China Agricultural University in years 2019, 2020, 2021 and 2022. 19C302, 19C304, 19C335, 19C355, 19C616, 19C705, 19C961, 19C1163, 19C1477, 19C1511 were provided by College of Horticulture, China Agricultural University.

Figure 1 Ripe fruits (A) and paraffin sections of anthesis ovaries (B–U) of 10 pepper accessions.

P-D, proximal-distal direction; M-L, medio-lateral direction. OML, OMW, and OWL are illustrated in (B).

Developmental analysis of fruit

Developmental analysis was performed on the fruits which were beared on the level IV branch points. Fruit maximal length (FML) and fruit maximal width (FMW) were measured from the anthesis stage to the breaker stage by using the slide caliper. In each accession, the measurements were performed on at least five flowers which were selected from 3–5 plants.

Histological analysis of the anthesis ovaries and mature fruits

Paraffin sections of the anthesis ovaries were made according to the protocol reported by Long (www.its.caltech.edu/~plantlab/protocols/insitu.pdf) with some modifications. Since the anthesis ovaries of some accessions were large, a small hole was punched on the wall of each ovary to promote the infiltration of FAA as well as other chemical agents. Images of each section were obtained using a microscope (Olympus BX53; Tokyo, Japan) coupled with a camera (Olympus DP72; Tokyo, Japan) under a ×20 objective lens. The entire ovary was reconstructed using overlapping images according to the method mentioned in Sun et al. (2015). In each accession, three sections were made from each ovary and at least five ovaries were collected from 3–5 plants. Free-hand sections of the fruits at breaker stage were made according to Sun et al. (2015). Images were taken with a microscope (Leica DFC450; Wetzlar, Germany) coupled with a digital camera (Olympus DP72; Tokyo, Japan). In each accession, three sections were made in each direction in each fruit and at least five fruits which were collected from 3–5 plants were investigated. Thickness of ovary wall and pericarp as well as cell number and size of the ovary wall and pericarp were investigated using ImageJ according to Sun et al. (2015).

RNA-Seq analysis

Anthesis ovaries of five representative accessions including 19C302, 19C335, 19C705, 19C961 and 19C1511 were collected for RNA-Seq analysis. Anthesis ovaries were dissected from five varieties and immediately frozen in liquid nitrogen. Library construction and sequencing were conducted by Biomarker Technologies Corporation (Beijing, China). Sequencing was performed on an Illumina novaseq 6000 platform (Illumina, San Diego, CA, USA). Illumina reads filtration, alignment and expression value calculation were all carried out on the BMKCloud (http://www.biocloud.net/). In the above-mentioned processes, Zunla1 V2.0 genome was selected as the reference genome and HISAT2 was employed to align the reads against the genome. The assembling of the aligned reads was conducted via StringTie. Differentially expressed genes (DEGs) were calculated using DESeq2 with the following parameters: the FPKM value of a gene in any of the five accessions was greater than or equal to two, FDR = 0.01, FC = 2. GO, KEGG and COG analyses, principal component analyses (PCA) of the transcript sequence variations and expression profile as well as the WGCNA analysis were all conducted on BMKCloud and the figures were modified using Photoshop CS5 (Adobe, San Jose, CA, USA). Heatmaps were generated using MeV4.9.0 software. Venn diagram analysis was conducted by VENNY2.1 (https://bioinfogp.cnb.csic.es/tools/venny/index.html). The raw sequencing data generated in this study are available in the NCBI Sequence Read Archive (PRJNA954557).

Identification of fruit shape related genes families in pepper

Members of the IQD, CaM, KLCR1, SPR2, OFP, PP2A, TON1 (TONNEAU1), TRM, WOX (WUSCHEL-related homeobox), MIKC (MADS, intervening keratin-like, and variable C terminal domains), CLV1, and CLV3 were identified in pepper genome database by BLAST using their homologs in Arabidopsis and tomato as queries. SMART (http://smart.embl-heidelberg.de/smart/batch.pl) and ExPASy (https://prosite.expasy.org/prosite.html) were used to analyze genes’ domains to eliminate the sequences without complete homologous domain. Physical location of each gene and chromosome length of each chromosome were obtained from the SGN (https://solgenomics.net/). Mapping of the genes to the chromosomes were conducted the Map Chart software 2.3.2 (Voorrips, 2002).

Results

Dynamic changes of fruit shape index in the studied pepper accessions

In order to clarify when the fruit shape of the studied accessions was determined, maximal length (OML/FML) and maximal width (OMW/FMW) of the anthesis ovary and mature fruit were measured during the fruit development and ripening, and anthesis ovary shape index (OSI = OML/OMW) as well as fruit shape index (FSI = FML/FMW) were calculated (Figs. S1A–S1J). 19C1477 had the highest FSI, following by 19C1511, 19C616, 19C705, 19C1163, 19C961, 19C335, 19C355, 19C304 and 19C302. Meanwhile, 19C1477 had the highest OSI, following by 19C1511, 19C616, 19C705, 19C961, 19C1163, 19C335, 19C355, 19C304 and 19C302, of which the last three accessions had the same OSI value. As to the overall variation trend, FSI of the flat-shaped fruits significantly decreased from anthesis to ~20 DPA (day post anthesis) (Figs. S1A, S1B). In contrast, FSI of the three elongated fruits, including the helical-, horn- and slender-shaped fruits, dramatically increased at the same time (Figs. S1E–S1J). In regard to the blocky-shaped fruits, FSI of 19C335 significantly increased until 20 DPA, however, FSI of 19C355 did not significantly change at the same time (Figs. S1C, S1D). After 20 DPA, FSI of all the accessions gradually stabilized (Figs. S1A–S1J). Among the studied accessions, the one with a higher FSI often showed a larger value of OSI, moreover, the Pearson correlation coefficient between FSI and OSI was 0.98, indicating there was a significant correlation between the two parameters.

Morphological and histological analyses of anthesis-stage ovaries and turning fruits

In the investigated anthesis ovaries, the largest values of OML and OWL (ovary wall length) were recorded in 19C616 and 19C355, respectively (Fig. 2A). Meanwhile, the largest values of OCN-PD (ovary wall cell number in proximal-distal direction), OCN-ML (ovary wall cell number in medio-lateral direction) and OMW were all discovered in 19C355 (Figs. 2A, 2B). In contrast, the largest value of OCS-PD (ovary wall cell size in proximal-distal direction) and largest value of OCS-ML (ovary wall cell size in medio-lateral direction) were found in 19C1163 and 19C705, respectively (Figs. 2A, 2B). As to the OWTK (ovary wall thickness) and OWCL (ovary wall cell layer), they were highly correlated in all the test accessions, except for 19C302 (Fig. 2C). Particularly, 19C705 had the largest OWTK, while, 19C302 had the largest OWCL (Fig. 2C).

Figure 2 Morphological and histological measurements of anthesis ovaries and turning fruits.

(A) Maximal length, wall length (numbers above or below the bars), cell number in proximal-distal (P-D) direction and average cell size in P-D direction of the anthesis ovaries. (B) Maximal width, cell number in medio-lateral (M-L) direction and average cell size in M-L direction of anthesis ovaries. (C) Ovary wall thickness and cell layer in the adaxial-abaxial direction of the anthesis ovaries. (D) Maximal length, cell number in P-D direction and average cell size in P-D direction of the turning fruits. (E) Maximal width, cell number in M-L direction and average cell size in M-L direction of the turning fruits. (F) Pericarp thickness and cell layer in adaxial-abaxial direction. Letters above or below the bars represent the significances of the HSD tests.

In regard to the turning fruits, accessions could be clearly classified into two groups based on the FML, FMW, PCN-PD (pericarp cell number in proximal-distal direction), PCN-ML (pericarp cell number in medio-lateral direction) as well as PTK (pericarp thickness) (Figs. 2D–2F). Specifically, 19C1477 had the largest FML (Fig. 2D), meanwhile, 19C355 had the largest FMW, the largest PCS-PD (pericarp cell size in proximal-distal direction) and the largest PCS-ML (pericarp cell size in medio-lateral direction) (Figs. 2D, 2E). As to 19C304, it had the largest PTK and the largest PCN-ML (Figs. 2E and F). For the largest PCN-PD and PCL (pericarp cell layer), they were observed in 19C616 and 19C302, respectively (Figs. 2D, 2F).

Correlation analysis indicated that, OML and OMW were significantly positively correlated with OCN-PD and OCN-ML, respectively (Table 1). As to the OWTK, it significantly positively correlated with OWCL, OCS-PD and OCS-ML (Table 1). In regard to the turning fruits, significantly positively correlation was observed between FML and PCN-PD (Table 1). Meanwhile, FMW was significantly positively correlated with PCN-ML and PCS-ML (Table 1). Similar to OWTK, PTK was significantly positively correlated with PCL, PCS-PD and PCS-ML (Table 1). In addition, PCN-ML was significantly positively correlated with PCS-ML (Table 1).

Table 1 Correlations of indices of anthesis ovaries and turning fruits.

Anthesis
ovary	OML	OWL	OMW	OWTK	OCN-PD	OCN-ML	
OCN-PD	0.68**	0.75**	--	--	1	--	
OCN-ML	--	--	0.94**	--	--	1	
OCS-PD	0.42	0.60*	--	0.65**	0.14	--	
OCS-ML	--	--	0.04	0.81**	--	−0.03	
OWCL	--	--	--	0.79**	--	--	
Turning
fruit	FML	FMW	PTK	PCN-PD	PCN-ML	
PCN-PD	0.86**	--	--	1	--	
PCN-ML	--	0.97**	--	--	1	
PCS-PD	−0.31	--	0.66**	−0.42	--	
PCS-ML	--	0.93**	0.91**	--	0.85**	
PCL	--	--	0.51*	--	--	
Note:

Abbreviations: OML, ovary maximal length; OWL, ovary wall length; OMW, ovary maximal width; OWTK, ovary wall thickness; OCN-PD, ovary wall cell number in P-D direction; OCN-ML, ovary wall cell number in M-L direction; OCS-PD, ovary wall cell size in P-D direction; OCS-ML, ovary wall cell size in M-L direction; FML, fruit maximal length; FMW, fruit maximal width; PTK, pericarp thickness; PCN-PD, pericarp cell number in P-D direction; PCN-ML, pericarp cell number in M-L direction; PCS-PD, pericarp cell size in P-D direction; PCS-ML, pericarp cell size in M-L direction; OWCL, ovary wall cell layer; PCL, pericarp cell layer. “--”, “*” and “**” indicate “not analyzed”, “0.01 ≤ p ≤ 0.05” and “p ≤ 0.01”.

Transcriptomic profile of the anthesis ovaries

For further illustrating the mechanism underlying different types of pepper fruit shape, RNA-seq was performed on the anthesis ovaries. The R package RNASeqPower was used to calculate the statistical power of this experiment. A power of 0.69 was found for genes that presented fold-change greater than two with a sample size of three (the number of biological replicates used in this study). Principal component analysis (PCA) of the mRNA sequence variation of the five representative accessions disclosed that samples collected from peppers bearing slender-shaped (19C1511), helical-shaped (19C705) and flat-shaped (19C302) fruits were separated from each other. In contrast, samples obtained from 19C355 and 19C961, which respectively beared blocky-shaped and horn-shaped fruits, were adjunct to each other but far from the three above-mentioned samples (Fig. 3A). In the aspect of the expression profiles, samples could not be clearly separated in the space defined by the PC1, PC2 and PC3, except for those of 19C1511, which was far from the others (Fig. 3B). Consistent with what is mentioned above, there were more DEGs in the comparisons involving 19C1511 (Fig. 3C). In each comparison, the number of the up-regulated genes were roughly equal to that of the down-regulated genes (Fig. 3C). The DEGs of each comparison were then input into the Clusters of Orthologous Groups (COG) analysis and the result showed that all the DEGs were enriched in 24 groups such as “general function prediction only”, “signal transduction mechanisms”, “cell wall/membrane/envelope biogenesis”, “cell cycle control, cell division, chromosome partitioning” and “cytoskeleton”, but were barely enriched in “cell motility” and “nuclear structure” (Fig. 3D). Additionally, the enrichment status of the DEGs obtained from different comparisons was not the same (Fig. 3D).

Figure 3 Overview of transcriptional analysis and comparison of helical and non-helical samples.

(A) Principal component analysis of the mRNA sequence variation of the five pepper accessions. (B) Principal component analysis of the gene expression profiles of all the samples. (C) DEG number between every two tested accessions. (D) COG enrichment analysis of all the comparisons. (E) Venn diagram of the four comparisons which were between the slender and non-slender samples. (F) Heatmap of the 592 DEGs shared by the four comparisons. (G) COG enrichment of the 592 DEGs shared by the four comparisons. (H) Interesting genes that were screened out of the 592 DEGs shared by the four comparisons. (I) Venn diagram of the four comparisons which were between the helical and non-helical samples. (J) Heatmap of the 135 DEGs shared by the four comparisons. (K) COG enrichment of the 135 DEGs shared by the four comparisons. (L) Interesting genes that were screened out of the 135 DEGs shared by the four comparisons.

Comparative transcriptomic analysis between slender-shaped and the other samples

As mentioned previously, the expression profile of the anthesis ovary of 19C1511 was different from that of other samples. In order to clarify this phenomenon, Venn diagram analysis was conducted among the four comparisons involving 19C1511 (Fig. 3E), and 592 common DEGs were identified which either extremely highly or lowly expressed in the anthesis ovary of 19C1511 (Fig. 3F). From the COG analysis, it could be known that those common DEGs were mainly enriched in “transcription”, “second metabolites biosynthesis, transport and catabolism”, “lipid transport and metabolism”, “cell cycle control, cell division, chromosome partitioning” and “cytoskeleton” (Fig. 3G). Later, 11 interested DEGs were identified, including two IQDs (IQD14: Capana03g003821, IQD25: Capana09g000006), two sugar transporter genes (SWEET1: Capana04g001460, SWEET10: Capana00g002537), four auxin-related genes (PCNT115: Capana03g004288, IAA16: Capana06g003073, IAA17: Capana03g000310, SAUR67: Capana00g002517), one gibberellin-related gene (GA2OX1: Capana00g003127) and two cell division and growth-related genes (EXO84A: Capana00g000539, NEDD1: Capana08g001023) (Fig. 3H). Among the above-mentioned genes, IQD14, PCNT115, IAA16, IAA17, SAUR67, GA2OX1 were upregulated in the anthesis ovary of 19C1511, while the left genes were down-regulated (Fig. 3H).

Comparative transcriptomic analysis between helical-shaped and non-helical-shaped samples

In order to further reveal the transcriptomic mechanisms underlying helical-shaped fruits, comparative transcriptomic analyses were conducted between the helical-shaped and non-helical-shaped anthesis ovaries, and meanwhile, Venn diagram analysis was performed among those four comparisons (Fig. 3I). A total of 135 common DEGs were identified which either extremely highly or lowly expressed in the anthesis ovary of 19C705 (Fig. 3J). COG analysis indicated that the above-mentioned DEGs were enriched in 16 function classes, such as “amino acid transport and metabolism”, “lipid transport and metabolism”, “cell wall/membrane/envelope biogenesis” and “cytoskeleton” (Fig. 3K). Furthermore, 10 interested DEGs were screened out, including a development-related gene MLP28, three cell proliferation/division-related genes (CYCA2;1, CYCA2;4 and ECA2), a cytoskeleton-related genes ATK4, and a sugar-related gene SWEET, four cell wall-related genes (CC1, LRX, FLA11 and PLL) (Fig. 3L). Among the above-mentioned genes, MLP28, CYCA2;4, ECA2, ATK4 and FLA11 were significantly higher expressed in the 19C705, while, the left genes were significantly lower expressed in the same sample (Fig. 3L).

Weighted gene correlation network analysis to explore genes responsive to the morphological changes of the anthesis ovaries

Weighted gene correlation network analysis (WGCNA) was conducted to explore the genes responsive to the changes of OSI, OML, OMW and OWTK, and 20 modules were identified (Fig. 4). Among those modules, the green one was significantly positively correlated with OMW, meanwhile, the orange one was significantly negatively correlated with OML (Fig. 4B). Additionally, two interested modules including the lighcyan and the black ones were also selected for further analyses, since the first one showed a higher correlation coefficient with OWTK and the second one was positively correlated with OSI but negatively correlated with OMW and OWTK (Fig. 4B).

Figure 4 Weighted gene correlation network analysis (WGCNA) of the RNA-seq data.

(A) Cluster dendrogram and modules. (B) Module-trait relationships. (C) Heatmap of genes of the black module. Number on the left indicates the gene number of the module. (D) COG enrichment analysis of genes in the black module. The class names have been shown in Fig. 4D. (E) KEGG enrichment of genes of the black module. (F) Interesting genes that were screened out of the black module. (G) Heatmap of genes of the green module. Number on the left indicates the gene number of the module. (H) COG enrichment analysis of genes in the green module. The class names have been shown in Fig. 4D. (I) KEGG enrichment of genes of the green module. (J) Interesting genes that were screened out of the green module. (K) Heatmaps of genes of the lightcyan and orange modules. Number above or below the heatmap indicate the gene number of each module. (L) KEGG enrichment of genes of the lightcyan and orange modules. (M) Interesting genes that were screened out of the lightcyan module. (N) Interesting genes that were screened out of the orange module.

The black module had the largest number of genes (435) among the four interested modules, which were either highly or lowly expressed in 19C1511 (Fig. 4C). Those genes were enriched in 21 function classes, such as “energy production and conversion”, “carbohydrate transport and metabolism”, “translation, ribosomal structure and biogenesis”, “transcription”, “signal transduction mechanisms” and “cytoskeleton” (Fig. 4D). Meanwhile, KEGG analysis showed that genes of this module were enriched in “phenylpropanoid biosynthesis”, “photosynthesis” and “oxidative phosphorylation” (Fig. 4E). Finally, 22 interested genes were identified according to the annotations, which either highly or lowly expressed in the anthesis ovary of 19C1511 (Fig. 4F). The above-mentioned genes consisted of an IQD14, three calcium-related genes (KIC, PBP1 and CaM), four auxin-related genes (AUX22D, IAA16/17, and GH3.6), two gibberellin-related genes (Snakin2 and GA2ox1), four transcription factor genes (AGL8, YABBY1 and two NACs), eight cell wall-related genes (CIF1, CSLG3, EXPL1, two GRPs, PME2.2 and two PGs) (Fig. 4F). Besides PME2.2 and two PGs, all those interested genes were highly expression in the anthesis ovary of 19C1511 (Fig. 4F).

The green module consisted of 192 genes which either obviously highly or lowly expressed in the anthesis ovary of 19C355 (Fig. 4G). Those genes were enriched in 18 function classes, such as “amino acid transport and metabolism”, “carbohydrate transport and metabolism”, “lipid transport and metabolism” and “signal transduction mechanisms” (Fig. 4H). Meanwhile, KEGG analysis revealed that the 192 genes were mainly enriched in “ABC transporters”, “tyrosine metabolism”, “fatty acid metabolism” and “biosynthesis of unsaturated fatty acids” (Fig. 4I). Further annotation analysis screened out 13 interested genes, including two auxin-related genes (GH3.1 and YUCCA3), two cytokinin-related genes (CYP735A1 and ZOG) and nine cell wall-related genes (two CELs, three EXTs, three PGs and a PLL19) (Fig. 4J). Besides YUCCA3, all those interested genes were highly expression in the anthesis ovary of 19C355 (Fig. 4J).

There were 87 and 37 genes in the lightcyan and orange modules, respectively (Fig. 4K). Genes of the lightcyan module either highly or lowly expressed in 19C302 and 19C705, meanwhile, genes of the orange module either highly or lowly expressed in 19C355 and 19C705 (Fig. 4K). Gene of the lightcyan and orange modules were enriched in “photosynthesis antenna proteins” and “endocytosis” in the KEGG analysis, respectively (Fig. 4L). Based on genes’ annotations, 11 interested genes were identified in the lightcyan module, including a transcription factor gene AGL31, a development-related gene MLPL34, three phytohormone-related genes (LOB4, EXL3 and UGT74E2) and six cell wall-related genes (PE, TBR, PGIP, RGXT2, EXT2 and GAE6) (Fig. 4M). As to the genes’ expression patterns in the anthesis ovaries, AGL31 was obviously relatively lowly expressed in 19C302 and 19C705; MLPL34 was relatively highly expressed in 19C705; LOB4 was relatively highly expressed in a 19C302; EXL3, PE and TBR were relatively highly expressed in 19C302 and 19C705; UGT74E2 was highly expressed in 19C302 and 19C705, but was lowly expressed in 19C355; PGIP was relatively highly expressed in 19C355 but was relatively lowly expressed in 19C302, meanwhile, GAE6 showed an opposite expression pattern; as to RGXT2, it was relatively highly expressed in 19C355 but was relatively lowly expressed in 19C705, of which the expression pattern was opposite to that of EXT2 (Fig. 4M). In the orange module, only six interested genes were discovered, including an auxin-related gene AXX15, two calcium-related genes (CMI1 and CML37) and three cell wall-related genes (GATL10, PGL3 and COBL4) (Fig. 4N). Expression of AXX15 and CMI1 was lower in the anthesis ovaries of 19C355 and 19C705; meanwhile, expression of CML37 and GATL10 was higher in 19C302 but lower in 19C355 and 19C705; expression of PGL3 was higher in 19C355; expression of COBL4 was higher in 19C705 but lower in 19C961 (Fig. 4N).

Fruit shape-related genes in pepper genome

In order to further explore the genes that were contributed to the pepper fruit changes, a total of 209 orthologs of tomato and pepper fruit shape-related genes were identified in pepper genome (Fig. 5). These fruit shape families are IQD (29), CaM (50), KLCR1 (5), SPR2 (5), AG (45), CLV1 (3), CLV3 (2), WUS (7), OFP (22), PP2A-A (2), PP2A-B (3), PP2A-C (11), TON1 (1) and TRM (24) (Fig. 5). Moreover, the 27 reported pepper fruit shape loci were also located on the chromosomes (Fig. 5). In another aspect, chromosome 3 (chr3) harbored the largest number (25) of the fruit shape-related genes, which was followed by chr2 (24 orthologs), chr01 (23 orthologs), chr10 (23 orthologs), chr09 (21 orthologs), chr11 (21 orthologs), chr08 (16 orthologs), chr04 (15 orthologs), chr06 (13 orthologs), chr05 (12 orthologs), chr07 (8 orthologs) and chr12 (8 orthologs) (Fig. 5).

Figure 5 Locations of the fruit shape related genes on pepper chromosomes.

The homologous genes in SUN pathway, LC-FAS pathway and OVATE pathway in pepper are represented by solid circle, square and triangle respectively; CAM-blue solid circle, IQD-orange solid circle, KLCR1-yellow solid circle, SPR2-red solid circle; AG-orange square, CLV1-yellow square, CLV3-green square, WUS-purple square; OFP-blue triangle, PP2A-A-yellow triangle, PP2A-B-green triangle, PP2A-C-red triangle, TON1-purple triangle, TRM-black triangle.

Discussion

Pepper fruit shape was determined during the developmental processes before and after anthesis

In this study, 10 pepper accessions bearing flat-, blocky-, horn-, helical- or slender-shaped fruits were investigated (Fig. 1A). Since significant differences of OSI had already been observed at anthesis stage, and meanwhile, FSI of 19C355 did not significantly change after anthesis (Figs. 1B–1U; S1), it can be presumed that FSI of the studied accessions was determined or at least partially determined during flower development before anthesis. Consistently, a significant difference of anthesis ovary shape was discovered between the pepper NILs carrying WT and fs10 locus (Borovsky et al., 2022), which further supported the above hypothesis. Moreover, OSI was significantly positively correlated with FSI, which further indicated that flower development played a critical role in the determination of pepper fruit shape. In another aspect, except for 19C355, FSI of all the investigated accessions obviously altered from 0 to 20 DPA, which resulted in the significant differences between OSI at anthesis stage and FSI at turning stage (Fig. S1), suggesting pepper FSI was also impacted by fruit development. Additionally, based on the developmental patterns as well as the positive correlation between the OSI and FSI (Fig. S1), it seemed that fruit developmental process enhanced the original patterns which were determined before anthesis, indicating, in each accession, fruit shape was determined by the developmental processes occurred both before and after anthesis. Taking together, it can be deduced that FSI of the studied pepper accessions was determined during flower and fruit developmental processes, of which the later process enhanced shape patterns set up in the earlier one. Similar phenomena were observed in tomato, for example, fruit shape of ovate or fs8.1 mutant, which respectively bears obovoid- and rectangular-shaped fruits, is predominantly determined before anthesis (Sun et al., 2015; Wu et al., 2015); in contrast, fruit shape of sun mutant, of which the fruit is elongated, is mainly formed before and a short time after anthesis (Xiao et al., 2008).

Histological bases of the OSI and FSI variations

Since the OML and OMW were significantly positively correlated with OCN-PD and OCN-ML, respectively (Table 1), it can be presumed that cell number alteration was the major reason for the OSI variation of the studied accessions. In contrast, as to the fruits at the turning stage, FML was significantly positively correlated with PCN-PD, whereas, FMW was significantly positively correlated with both PCN-ML and PCS-ML (Table 1), suggesting FSI is determined by both cell division and cell expansion in the investigated accessions. In regard to the OWTK and PTK, based on the correlations (Table 1), it can be deduced that they were regulated by both cell division in the abaxial-adaxial direction and cell expansion in the P-D (proximal-distal) and M-L (medio-lateral) directions. The above-mentioned developmental patterns were not similar to those of sun and ovate mutations in tomato, because those two loci are cell number regulators and do not obviously impact cell size (Wu et al., 2011, 2018). In another aspect, a negative correlation was discovered between PCN-PD and PCS-PD, and similar phenomenon has also been observed in fs8.1 NIL, in which the pericarp cell size decreased with the increase of cell number (Sun et al., 2015). However, in the M-L direction, PCN-ML was significantly positively correlated with PCS-ML, which has never been discovered in any of tomato fruit shape mutants, suggesting a different fruit shape regulatory pattern may exist in pepper. Therefore, fruit shape of the 10 pepper accessions was controlled by both cell division and cell expansion, of which the regulator mechanism may be different from that of tomato.

Transcriptomic regulatory mechanisms underlying different shapes of pepper fruits

Since anthesis is a critical stage for pepper fruit shape determination, transcriptomic analysis was conducted on the anthesis ovaries. From the mRNA sequence variation, gene expression profiles as well as the DEG number between every two accessions (Figs. 3A–3C), it can be deduced that 19C1511, the accession bearing slender-shaped fruits, may have a distinct transcriptome than the accessions bearing wider fruits. A recent variome study indicated that some elongated peppers may have different domestication experiences from that of the blocky-shaped peppers, and a TRM25 gene (Capana03g002426) was identified to underly the fsi locus, which controls fruit elongation (Cao et al., 2022b). Meanwhile, in another study, an OFP20 gene (Capana10g001230) has been confirmed to underly fs10, and the lower expression level of OFP20, which was caused by a 42-bp deletion in the upstream region of this gene, resulted in the elongation of anthesis ovary and fruit (Borovsky et al., 2022). Therefore, it can be presumed that the OFP-TRM pathway plays an important role in the regulation of pepper fruit shape. Correspondingly, in this study, expression of some OFPs, such as Capana04g00381 and Capana09g001195, showed decreased trends when comparing with that of other accessions (Fig. 5), suggesting a potential role of the OFP-TRM pathway in the regulation of the slender-shaped fruit in this study. In another aspect, an IQD14 gene (Capana03g003821) and a CaM gene (Capana10g001655) were discovered to be highly expressed only in the anthesis ovary of the slender-shaped accession 19C1511 (Figs. 4F, 5). The IQD gene family harbors an important fruit shape gene SUN, and the higher expression of SUN as well as its orthologs led to dramatic elongation of ovaries and fruits in different plants (Xiao et al., 2008; Wu et al., 2011; Jin et al., 2018; Li et al., 2022; Ma et al., 2022a). At the protein level, IQD directly interacts with CaM (Bürstenbinder et al., 2013). Thus, it can be deduced that the IQD-CaM pathway may take part in the regulation of pepper fruit shape. Moreover, CaM was also detected to be lowly expressed in the anthesis ovary of 19C302 (Fig. 4F), an accession carrying flat-shaped fruits, which further confirmed its potential role in the control of pepper fruit elongation. In addition, two sugar transporter genes (Capana04g001460 and Capana00g002537) and two cell division and growth-related genes (Capana00g000539 and Capana08g001023) were all down-regulated in the ovary of 19C1511 (Fig. 3H), which may partially explain the phenomenon that the ovary of this accessions was smaller than those of others.

As to the 19C355, it had the largest ovary wall cell number in both P-D and M-L directions among all the accessions (Figs. 2A and 2B), which can be attributed to the higher expression of CYP735A1 (Fig. 4J). In Arabidopsis, CYP735A1 encodes a cytokinin hydroxylase that catalyzes the biosynthesis of trans-zeatin (Takei, Yamaya & Sakakibara, 2004), and the higher expression of this gene may lead to the increased cytokinin level and may finally result in the cell number increase in anthesis ovary. In addition, a ZOG gene was also highly expressed in the anthesis ovary (Fig. 4J). This gene encodes a cytokinin O-glucosyltransferase which is involved in the homeostasis of this hormone by catalyzing the reversible inactivation of cytokinin (Hluska, Hlusková & Emery, 2021), and the higher expression of ZOG here may be due to the feedback loop. In contrast with cytokinin, auxin level did not seem to play a critical role in the regulation of 19C355 ovary shape, because GH3.1, a gene involved in auxin degradation, was highly expressed in anthesis ovary, meanwhile, YUCCA3, a gene involved in auxin biosynthesis, was lowly expressed in the same organ (Fig. 4J), which indicated a decreased auxin level.

Although helical growth patterns are common in plants, it is still rare to see helical-shaped fruits in horticultural crops. The spiral growth of the plant organs is mainly due to the mutations of the cytoskeleton-related proteins, such as a-tubulin, β-tubulin, MAP and kinesin, meanwhile, mutations of IQD67 and calmodulin-like proteins could also lead to this phenotype (Buschmann & Borchers, 2020). Additionally, cellulose microfibril alignment is also involved in the spiral growth (Baskin, 2005). Consistently, in this study, a kinesin gene, ATK4, was identified to be significantly highly expressed in the anthesis ovary of 19C705, the accession bearing helical-shaped fruits, meanwhile, a cellulose biosynthesis-related gene, CC1, encoding COMPANION OF CELLULOSE SYNTHASE 1, was observed to be significantly lowly expressed in the same sample (Fig. 3L), suggesting the cytoskeleton and cellulose biosynthesis may be impacted in 19C705. Besides CC1, three other cell wall-related genes, including LRX, FLA11 and PLL, also differently expressed among the studied accessions (Fig. 3L). LRX encodes a cell wall-attached extracellular regulator which takes part in cell wall formation and cell growth (Herger et al., 2020). FLA11 encodes a Fasciclin-Like Arabinogalactan protein which plays a part in second cell wall development and the balance of lignin and cellulose synthesis/deposition (Ma et al., 2022b). PLL encodes a pectin lyase-like superfamily protein and is involved in cell wall dynamics (Marín-Rodríguez, Orchard & Seymour, 2002). Thus, the different expression of the above-mentioned three genes further supported the hypothesis that cell wall-related pathways may be involved in the helical growth of pepper fruits. In another aspect, cell proliferation may also take part in the regulation of the helical growth, since CYCA2;1, CYCA2;4 and ECA2 were either significantly higher or lower expressed in 19C705 when compared with other accessions (Fig. 3L). In Arabidopsis, CYCA2;1 and CYCA2;4 are involved in the cell proliferation in vein (Donner & Scarpella, 2013). As to ECA2, it encodes an epsin-like clathrin adaptor and its homologous proteins in Arabidopsis localizes to the growing cell plate in cells undergoing cytokinesis and are involved in cell division (Song et al., 2012). Consistently, based on our previous study, the fruit helical growth pattern in 19C705 is due to the inclined arrangement of the pericarp cells (Cao et al., 2022a), which further suggested that the helical growth of 19C705 may involve the alteration of cell division pattern. In addition, MLP28 gene, encoding Major Latex Protein 28, was significantly higher expressed in 19C705 than other accessions (Fig. 3L). In Arabidopsis, MLP28 plays a part in the regulation of leaf morphology and patterning, and down-regulation of this gene led to elongated petioles and alterations in leaf curvature (Litholdo et al., 2016), suggesting the possibility of this gene in the regulation of the helical growth of pepper fruit. Therefore, cell wall-, cell proliferation/division-related pathways as well as MLP28 gene may be involve in the helical growth of pepper. Interestingly, in our previous study, a comparison of the transcriptomes of the anthesis ovaries of 19C705 and 19C961 identified a set of phytohormone-related genes (Cao et al., 2022a), which were never found in the multiple comparisons in this study, indicating the phytohormone-related pathway may not be the key factor in the regulation of helical growth of 19C705.

Conclusions

Fruit shape of the studied pepper accessions was formed during the flower and fruit developmental processes before and after anthesis, with anthesis being a pivotal stage for the determination of fruit shape. Ovary shape index variations of the studied accessions were mainly attributed to the cell number change, while, fruit shape index changes were predominantly caused by the alterations of cell division and cell expansion. The thickness of the ovary wall and the pericarp were significantly influenced by cell division along the abaxial-adaxial axis and cell expansion in the proximal-distal and medio-lateral directions. At the transcriptional level, genes related to development, cell proliferation/division, the cytoskeleton, and the cell wall may contribute to the regulation of helical growth in peppers. At the same time, the OFP-TRM and IQD-CaM pathways are highly likely to be involved in regulating the formation of the slender-shaped fruit investigated in this study.

Supplemental Information

Supplemental Information 1 Dynamic changes of fruit shape index (FSI) of accessions 19C302 (A), 19C304 (B), 19C335 (C), 19C355 (D), 19C616 (E), 19C705 (F), 19C961 (G), 19C163 (H), 19C1477 (I) and 19C1511 (J).

In each panel, the number in front of the slash indicates the average value of the anthesis ovary shape index, meanwhile, the number behind the slash represents the average value of fruit shape index at the last development stage. Lowercases and capital letters indicate the significance of the HSD test of the anthesis ovary shape index and fruit shape index at the last development stage, respectively. “**” indicates the significant difference between the anthesis ovary shape index and fruit shape index at the last development stage at 0.01 level.

Additional Information and Declarations

Competing Interests

Author Contributions

Data Availability

The authors declare that they have no competing interests.

Yixin Wang performed the experiments, analyzed the data, prepared figures and/or tables, authored or reviewed drafts of the article, and approved the final draft.

Shijie Ma performed the experiments, analyzed the data, prepared figures and/or tables, authored or reviewed drafts of the article, and approved the final draft.

Xiaomeng Cao performed the experiments, analyzed the data, authored or reviewed drafts of the article, and approved the final draft.

Zixiong Li performed the experiments, analyzed the data, authored or reviewed drafts of the article, and approved the final draft.

Bingqing Pan performed the experiments, authored or reviewed drafts of the article, and approved the final draft.

Yingying Song performed the experiments, authored or reviewed drafts of the article, and approved the final draft.

Qian Wang conceived and designed the experiments, prepared figures and/or tables, and approved the final draft.

Huolin Shen conceived and designed the experiments, prepared figures and/or tables, and approved the final draft.

Liang Sun conceived and designed the experiments, prepared figures and/or tables, authored or reviewed drafts of the article, and approved the final draft.

The following information was supplied regarding data availability:

The raw sequencing data generated in this study are available at the Sequence Read Archive (SRA) of the National Center for Biotechnology Information (NCBI): PRJNA954557.

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
