# Peer review of "Morphological, histological and transcriptomic mechanisms underlying different fruit shapes in Capsicum spp"

_PeerJ, doi:10.7717/peerj.17909_

## Round 0.1 · original submission · Major Revisions

Dear Dr. Sun,

I have now received the reviewer's comments (below). The reviewers found the manuscript descriptive, but it does add new information to the field. Therefore, I recommend the resubmission of a new version of the manuscript given that you address every comment.

Best regards

Fabio Nogueira

·

Basic reporting

The authors propose identifying gene expression profiles from genes controlling the main types of fruit shape that may be related to some of the recent QTLs identified in the pepper genome. Indeed, this is unknown for this crop compared to the main known QTLs identified in tomato for the same trait. The authors succeeded in this main aim since they have found 209 orthologs of tomato contributing to pepper fruit changes. Moreover, this study correlates the gene expression signatures of these genes with the main types of pepper fruits. This work provides new insights into further studies aiming to unveil the molecular mechanisms of fruit morphogenesis.

Experimental design

The experimental design is appropriate to achieve the aims. The proposed morphological measurement parameters (ovary and fruit index) are conducted in the proper way, as well as the data from RNAseq. These methods provide statistically robust data to achieve the proposed goals.

Validity of the findings

The findings are based and robust data, especially regarding the RNAseq analyses.

Additional comments

I recommend following the suggestion related to the choice of the main and supplementary figures, as well reviewing the highlighted misconceptions.

·

Basic reporting

The manuscript of Wang et al. rigorously describes the fruit formation for ten Capsicum cultivars belonged to five clearly distinctive fruit shapes. Then, they make a comprehensive transcriptome analysis with five of them from tissue collected at anthesis stage. The introduction or background is extensively focused on tomato fruit shape genes known; however, is not mentioned the recently cloned gene Globe (see https://doi.org/10.1038/s41438-021-00574-3).
The article is well-structured, and the results are supported by clear figures, tables and raw data. The authors have to carefully revise the use of genes and abbreviations because several times are not extensible written the first time used or are not consistently mentioned throughout the manuscript. For example: FML, FMW, WOX, MICK, CAM or CaM. Others are mentioned differently in text and figures, i.e, OWTK in Fig 5B.

Since the work intent to compare fruit development of ten cultivars selected for their discrepancy in fruit shape, I suggest making the figure 2 with mean values of the FSI in the same panel showing the differences among all the cultivars (same y axe). Errors bars can be help to graphically display the significant differences among means values.

The "Morphological and histological analysis of anthesis..." section is hard to read and follow. The authors just have to explain the significant differences or extreme differences between the phenotypes, or only describes those more significant for the biological process under study.

The supplementary figure needs a Figure caption.

Experimental design

The objective of the research is clear, the selected methods and plant material are appropriate, and the investigation was performed under rigorous procedures and protocols.
The study is according the aims and scope of the journal, and it will be of interest for Plant Biologist in general and those involved in fruit and development process in particular.

Validity of the findings

The results of this work are descriptive and original in regard with morphological and gene expression variation related to distinctively fruit shapes in Capsicum. The authors made a comprehensive study about the fruit development on ten cultivars of capsicum from anthesis to full size.
The selected time point for the transcriptome analysis is adequate; however, the majors genes underlying to change the fruit morphology in Capsicum likely are not differentially expressed in anthesis. The plant material also is not the most appropriate to directly associate fruit phenotype with underlying genes. The cultivars are securely different for many others fruit attributes than the shape. The results had had more relevance if the authors had compared the transcriptome of elongated versus not elongated fruit instead of pair comparisons.
Based on the previous statement, I suggest rewriting the conclusion to be less speculative and limited to the supporting results.

Additional comments

Under the above considerations, I think the paper in its present format is not fully adequate for acceptation, but after major revision can achieve the standard requested by the journal.

---

## Round 0.2 · accepted · Accept

Dear Dr. Sun,

Your manuscript "Morphological, histological and transcriptomic mechanisms underlying different fruit shapes in Capsicum spp."has now been evaluated by two experts in the field, and you have addressed all the reviewers' comments.

Thus, I am please to inform that your manuscript has been accepted for publication in PeerJ.

Your article will be checked and you will receive a list of production tasks in approximately five business days. After you complete these tasks, your proofing PDF will be created (please do not proof check your reviewing PDF!).

Congratulations and my best regards

Assistant editor
Fabio Nogueira


·

Basic reporting

no comment

Experimental design

no comment

Validity of the findings

no comment

Additional comments

The authors properly addressed my main concerns and suggestions. I am particularly pleased about transferring the figure "Locations of the fruit shape related genes on pepper chromosomes" from supplemental material to the main results. In addition, some inappropriate conclusions were also corrected (e.g., genes of the OFP-TRM pathways), and deeper analyses of the DEGs from the elongated pepper accession (19C1511) were included. Thus, I consider this reviewed version of the manuscript appropriate to be published.

·

Basic reporting

The manuscript by Wang et al. rigorously describes fruit formation in ten Capsicum cultivars, which belong to five clearly distinctive fruit shapes. Additionally, the authors conduct a comprehensive transcriptome analysis on five of these cultivars using tissue collected at the anthesis stage. This study serves as a framework for exploring the diversity in fruit shape and their underlying transcriptome profiles in pepper.

Experimental design

As previously mentioned, the research objectives are clearly defined, the selected methods and plant materials are appropriate, and the investigation was conducted with rigorous procedures and protocols. The study aligns with the aims and scope of the journal and will be of interest to plant biologists in general, especially those studying fruit development processes.

Validity of the findings

The findings of this study provide descriptive and original insights into the morphological and gene expression variations associated with distinct fruit shapes in Capsicum. The authors conducted a comprehensive and rigorous investigation of fruit development across ten Capsicum cultivars, spanning from anthesis to full size. The results demonstrate originality and constitute a robust scientific study.

Additional comments

The authors have responded and incorporated most of the suggested changes. In summary, I believe that this version should be accepted for publication